# LEARNING WITH A MOLE: TRANSFERABLE LATENT SPATIAL REPRESENTATIONS FOR NAVIGATION WITHOUT RECONSTRUCTION

**Guillaume Bono, Leonid Antsfeld, Assem Sadek, Gianluca Monaci & Christian Wolf**
Naver Labs Europe Meylan, France
`{firstname.lastname}@naverlabs.com`

## ABSTRACT

Agents navigating in 3D environments require some form of memory, which should hold a compact and actionable representation of the history of observations useful for decision taking and planning. In most end-to-end learning approaches the representation is latent and usually does not have a clearly defined interpretation, whereas classical robotics addresses this with scene reconstruction resulting in some form of map, usually estimated with geometry and sensor models and/or learning. In this work we propose to learn an actionable representation of the scene independently of the targeted downstream task and without explicitly optimizing reconstruction. The learned representation is optimized by a blind auxiliary agent trained to navigate with it on multiple short sub episodes branching out from a waypoint and, most importantly, without any direct visual observation. We argue and show that the blindness property is important and forces the (trained) latent representation to be the only means for planning. With probing experiments we show that the learned representation optimizes navigability and not reconstruction. On downstream tasks we show that it is robust to changes in distribution, in particular the sim2real gap, which we evaluate with a real physical robot in a real office building, significantly improving performance.

## 1 INTRODUCTION

Navigation in 3D environments requires agents to build actionable representations, which we define as in Ghosh et al. (2019) as "*aim(ing) to capture those factors of variation that are important for decision making*". Classically, this has been approached by integrating localization and reconstruction through SLAM (Thrun et al., 2005; Bresson et al., 2017; Lluvia et al., 2021), followed by planning on these representations. On the other end of the spectrum we can find end-to-end approaches, which map raw sensor data through latent representations directly to actions and are typically trained large-scale in simulations from reward (Mirowski et al., 2017; Jaderberg et al., 2017) or with imitation learning (Ding et al., 2019). Even for tasks with low semantics like *PointGoal*, it is not completely clear whether an optimal representation should be "handcrafted" or learned. While trained agents can achieve extremely high success rates of up to 99% (Wijmans et al., 2019; Partsey et al., 2022), this has been reported in simulation. Performance in real environments is far lower, and classical navigation stacks remain competitive in these settings (Sadek et al., 2022). This raises the important question of whether robust and actionable representations should be based on precise reconstruction, and we argue that an excess amount of precision can potentially lead to a higher internal sim2real gap and hurt transfer, similar (but not identical) to the effect of fidelity in training in simulation (Truong et al., 2022). Interestingly, research in psychology has shown that human vision has not been optimized for high-fidelity 3D reconstruction, but for the usefulness and survival (Prakash et al., 2021).

We argue that artificial agents should follow a similar strategy and we propose tailored auxiliary losses, which are based on interactions with the environment and directly target the main desired property of a latent representation: its *usability* for navigation. This goal is related to *Cognitive Maps*, spatial representations built by biological agents known to emerge from interactions (Tolman, 1948; Blodgett, 1929; Menzel, 1973), even in blind agents, biological (Lumelsky & Stepanov, 1987) or artificial ones (Wijmans, 2022; Wijmans et al., 2023).

Inspired by this line of research, we propose learning a latent spatial representation we call *Navigability* and avoid spending training signals on learning to explicitly reconstruct the scene in unnecessary detail, potentially not useful or even harmful for transfer. We augment the amount of information carried by the training signal compared to reward. We consider the ability of performing local navigation an essential skill for a robot, i.e. the capability to detect free navigable space, avoid obstacles, and find openings in closed spaces in order to leave them. We propose to learn a representation which optimizes these skills directly, prioritizing usability over fidelity, as shown in Figure 1. A representation is built by an agent through sequential integration of visual observations. This representation is passed to a *blind* auxiliary agent, which is trained to use it as its sole information to navigate to a batch of intermediate subgoals. Optimizing over the success of the blind auxiliary agent leads to an actionable representation and can be done independently of the downstream task.

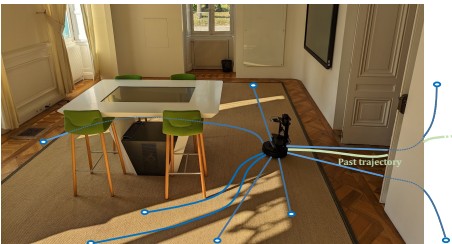

Figure 1: Without reconstruction, we learn an actionable map-like representation computed by an agent of the visual observations collected along its trajectory ▬. We optimize for its *usefulness*: a representation estimated at point ♩ is passed to a blind auxiliary agent trained to reach subgoals ⊙ on short episodes ▬. Solving this requires *sufficient latent* reconstruction of the scene, and we show that blindness of the auxiliary agent is a key property. We train in simulation and transfer to a real environment.

We explore the following questions: (i) Can a latent cognitive map be learned by an agent through the communication with a blind auxiliary agent? (ii) What kind of spatial information does it contain? (iii) Can it be used for downstream tasks? (iv) Is it more transferable than end-to-end training in out-of-distribution situations such as sim2real transfer?

## 2    RELATED WORK

**Navigation with mapping and planning** — Classical methods typically require a map (Burgard et al., 1998; Marder-Eppstein et al., 2010; Macenski et al., 2020) and are composed of three modules: mapping and localization using visual observations or Lidar (Thrun et al., 2005; Labbé & Michaud, 2019; Bresson et al., 2017; Lluvia et al., 2021), high-level planning (Konolige, 2000; Sethian, 1996) and low-level path planning (Fox et al., 1997; Rösmann et al., 2015). These methods depend on sophisticated noise filtering, temporal integration with precise odometry and loop closure. In comparison, we avoid the intermediate goal of explicit reconstruction, directly optimizing *usefulness*.

**End-to-end training** — on the other side of the spectrum we find methods which directly map sensor input to actions and trained latent representations, either flat vectorial like GRU memory, or structured variants: neural metric maps encoding occupancy (Chaplot et al., 2020b), semantics (Chaplot et al., 2020a) or fully latent metric representations (Parisotto & Salakhutdinov, 2018; Beeching et al., 2020b; Henriques & Vedaldi, 2018); neural topological maps (Beeching et al., 2020a; Shah & Levine, 2022; Shah et al., 2021); transformers (Vaswani et al., 2017) adapted to navigation (Fang et al., 2019; Du et al., 2021; Chen et al., 2022; Reed et al., 2022); and implicit representations (Marza et al., 2023). While these methods share our goal of learning *useful* and actionable representations, these representations are tied to the actual downstream task, whereas our proposed "Navigability" optimizes for local navigation, an important capability common to all navigation tasks.

**Pre-text tasks** — Unsupervised learning and auxiliary tasks share a similar high-level goal with our work, they provide a richer and more direct signal for representation learning. Potential fields (Ramakrishnan et al., 2022) are trained from top down maps and contain unexplored areas and estimates of likely object positions. A similar probabilistic approach has been proposed in RECON by Shah et al. (2021). In (Marza et al., 2022), goal direction is directly supervised. Our work can be seen as a pre-text task directly targeting the usefulness of the representation combined with an inductive bias in the form of the blind auxiliary agent.

**Backpropagating through planning** — in this line of work a downstream objective is backpropagated through a differentiable planning module to learn the upstream representation. In (Weerakoon et al., 2022) and similarly in (Dashora et al., 2021), this is a cost-map used by a classical planner. *Neural-A\** (Yonetani et al., 2021) learns a cost-map used by a differentiable version of *A\**. Similarly, *Cognitive Mapping and Planning* (Gupta et al., 2017) learns a mapping function by backpropagating

through *Value Iteration Networks* (Tamar et al., 2016). Our work shares the overall objective, the difference being that (i) we optimize *Navigability* as an additional pre-text task and, (ii) we introduce the blind agent as inductive bias, which minimizes the reactive component of the task and strengthens the more useful temporal integration of a longer history of observations — see Section 4. Somewhat related to our work, motion planning performance has been proposed as learnable evaluation metric (Philion et al., 2020), and attempts have been made to leverage this kind of metric for learning representations (Philion & Fidler, 2020; Zeng et al., 2021), albeit without application to navigation.

**Sim2Real transfer** — transferring representations to real world has gained traction since the recent trend to large-scale learning in simulated 3D environments (Höfer et al., 2020; Chattopadhyay et al., 2021; Kadian et al., 2020; Anderson et al., 2018; Dey et al., 2023). Domain randomization randomizes the factors of variation during training (Peng et al., 2018; Tan et al., 2018), whereas domain adaptation transfers the model to real environments, or in both directions (Truong et al., 2021), through adversarial learning (Zhang et al., 2019), targeting dynamics (Eysenbach et al., 2021) or perception (Zhu et al., 2019), or by fine-tuning to the target environment (Sadek et al., 2022). Table 6 in appendix lists some efforts. Our work targets the transfer of a representation by optimizing its usefulness instead of reconstruction.

**Biological agents** — like rats, have been shown to build *Cognitive Maps*, spatial representations emerging from interactions (Tolman, 1948), shown to partially emerge even when there is no reward nor incentives (Blodgett, 1929; Tolman, 1948). Similarly, chimpanzees are capable of developing greedy search after interactions (Menzel, 1973). Blind biological agents have been shown to be able navigate (Lumelsky & Stepanov, 1987), which has recently been replicated for artificial agents (Wijmans et al., 2023). Certain biological agents have also been shown to develop grid, border and place cells (Hafting et al., 2005), which have also been reproduced in artificial agents (Cueva & Wei, 2018; Banino et al., 2018). There are direct connections to our work, where a representation emerges from interactions of a blind agent.

**Goal-oriented models** — are learned through optimizing an objective with respect to subgoals. *Hindsight Experience Replay* (Andrychowicz et al., 2017) optimizes sample efficiency by reusing unsuccessful trajectories, recasting them as successful ones wrt. different goals. Chebotar et al. (2021) learn "*Actionable Models*" by choosing different states in trajectories as subgoals through hindsight replay. In Reinforcement Learning (RL), successor states provide a well founded framework for goal dependent value functions (Blier et al., 2021). Subgoals have also recently been integrated into the MDP-option framework (Lo et al., 2022). Similarly, subgoals play a major part in our work.

## 3    LEARNING NAVIGABILITY

We learn a representation useful for different visual navigation problems, and without loss of generality we formalize the problem as a *PointGoal* task: An agent receives RGB-D observations $\mathbf{o}_t$ and a Euclidean goal vector (*GPS+Compass*) $G_t$ at each time step $t$ and must take actions $\mathbf{a}_t$, which are typically discrete actions from a given alphabet {FORWARD 25cm, TURN_LEFT 10°, TURN_RIGHT 10° and STOP}. The agent sequentially builds a representation $\mathbf{r}_t$ from the sequence of observations $\{\mathbf{o}_{t'}\}_{t'<t}$ and the previous action $\mathbf{a}_{t-1}$, and a policy $\pi$ predicts a distribution over actions,

$$\mathbf{r}_t = f(\mathbf{o}_t, G_t, \mathbf{r}_{t-1}, \mathbf{a}_{t-1}) , \ p(\mathbf{a}_t) = \pi(\mathbf{r}_t), \tag{1}$$

where $f$ in our case is a neural network with GRU memory (Cho et al., 2014), but which can also be modeled as self-attention over time as in (Chen et al., 2021; Janner et al., 2021; Reed et al., 2022). We omit dependencies on parameters and gating mechanisms from our notations.

We do not focus on learning the main policy $\pi$, which can be trained with RL, imitation learning (IL) or other losses on a given downstream task. Instead, we address the problem of learning the representation $\mathbf{r}_t$ through its *usefulness*: we optimize the amount of information $\mathbf{r}_t$ carries about the *navigability* of (and towards) different parts of the scene. When given to a blind auxiliary agent, this information should allow the agent to navigate to any sufficiently close point, without requiring any

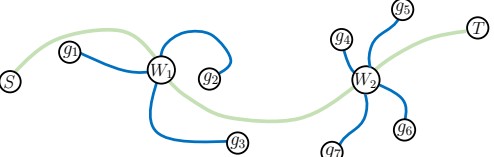

Figure 2: Our training data is organized into "long episodes" from start $S$ to target $T$ navigated by the main agent $\pi$ and "short episodes" branching out at waypoints $W_i$ to subgoals $g_j$.

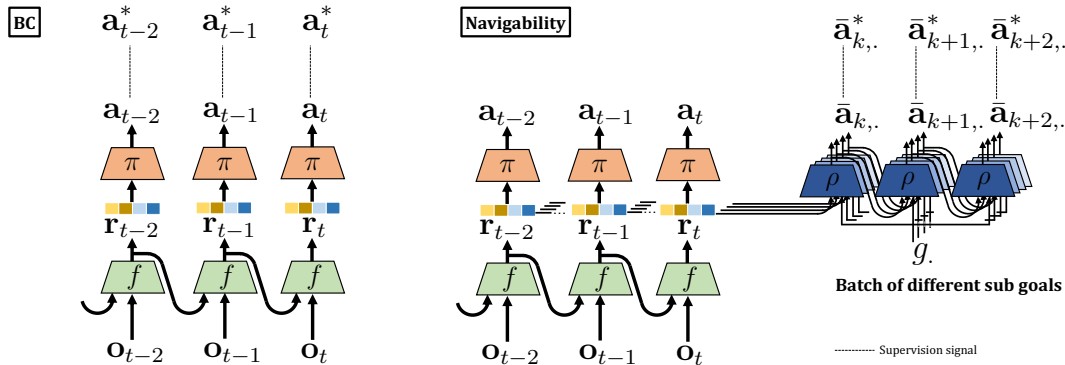

Figure 3: **Learning navigability:** the difference between classical behavior cloning (Left), which directly learns the main target policy $\pi$ on a downstream task, and learning navigability (Right), i.e. learning a representation $\mathbf{r}_t$ predicted by $f$ through a blind auxiliary policy $\rho$ allowing to navigate by predicting sequences of actions for a batch of different subgoals for each time instant $t$. The representation is then used by the downstream policy $\pi$.

visual observation during its own steps. We cast this as a tailored variant of behavior cloning (BC) using privileged information from the simulator.

We organize the training data into episodes, for each of which we suppose the existence of optimal (shortest) paths, e.g. calculated in simulation from GT maps. We distinguish between long and short episodes, as shown in Figure 2. During training, long episodes are followed by the (main) agent $\pi$, which integrates observations into representations $\mathbf{r}_t$ as in eq. (1). At waypoints $W_i$ sampled regularly during the episode, representations $\mathbf{r}_t$ are collected and sent to a batch (of size $B$) of multiple blind auxiliary agents, which branch out and are trained to navigate to a batch of subgoals $\{g_j\}_{j=1...B}$.

The blind agent is governed by an auxiliary policy $\rho$ operating on its own recurrent GRU memory $\mathbf{h}$,

$$\mathbf{h}_{k,j} = \bar{f}(g_j, \mathbf{h}_{k-1,j}, \mathbf{r}_t, \bar{\mathbf{a}}_{k-1,j}), \tag{2}$$

$$p(\bar{\mathbf{a}}_{k,j}) = \rho(\mathbf{h}_{k,j}), \tag{3}$$

where the index $j$ is over the subgoals of the batch, $k$ goes over the steps of the short episode, and the actions $\bar{\mathbf{a}}_{k,j}$ are actions of the aux agent. The representation $\mathbf{r}_t$ collected at step $t$ of the main policy remains constant over the steps $k$ of the auxiliary policy, with $\bar{f}$ its GRU update function. This is illustrated in Figure 3 ($\bar{f}$ not shown, integrated into $\rho$).

We train the policy $\rho$ and the function $f$ predicting the representation $\mathbf{r}_t$ jointly by BC minimizing the error between actions $\bar{\mathbf{a}}_{k,j}$ and the GT actions $\bar{\mathbf{a}}_{k,j}^*$ derived from shortest path calculations:

$$\hat{f} = \arg\min_{f,\rho} \sum_k \sum_{j=1}^{B} \mathcal{L}_{CE}(\bar{\mathbf{a}}_{k,j}, \bar{\mathbf{a}}_{k,j}^*), \tag{4}$$

where $\mathcal{L}_{CE}$ is the cross-entropy loss and the index $k$ runs over all steps in the training set. Let us recall again a detail, which might be slightly hidden in the notation in equation (4): while the loss runs over the steps $k$ in short-trajectories, these steps are attached to the steps $t$ in long episodes through the visual representation $\mathbf{r}_t$ built by the encoder $f$, as in equation (3). The auxiliary policy $\rho$ is a pre-text task and not used after training. Navigation is performed by the main agent $\pi$, which is finetuned on its own downstream objective.

**Navigability vs. BC** — there is a crucial difference to classical Behavior Cloning, which trains the main policy jointly with the representation $f$ from expert trajectories mimicking or approximating the desired optimal policy (see Ramrakhya et al. (2023) for comparisons), i.e.:

$$\mathbf{r}_t = f(\mathbf{o}_t, G_t, \mathbf{r}_{t-1}, \mathbf{a}_{t-1}) \ , \ p(\mathbf{a}_t) = \pi(\mathbf{r}_t) \tag{5}$$

$$(\hat{f}, \hat{\pi}) = \arg\min_{f,\pi} \sum_{i \in \mathcal{D}} \mathcal{L}_{CE}(\mathbf{a}_i^*, \pi(\mathbf{r}_i)), \tag{6}$$

where $G_t$ is the (global) navigation goal and $\mathcal{D}$ the training data. In the case of navigation, these experts are often shortest path calculations or human controllers combined with goal selection through hindsight. It is a well-known that BC is suboptimal for several reasons (Kumar et al., 2022). Amongst others, it depends on sufficient sampling of the state space in training trajectories, and it fails to adequately learn exploration in the case where no single optimal solution is available to the agent due to partial observability. In contrast, our navigability loss trains the representation $\mathbf{r}_t$ only, and can be combined with independently chosen downstream policies.

**Subgoal mining** — For a given compute budget, the question arises how many steps are spent on long vs. short episodes, as each step spent on a short episode is removing one visual observation from training — $\rho$ is blind. We sample waypoints on each long episode and attach a fixed number of subgoals to each waypoint sampled uniformly at a given Euclidean distance. Mining subgoal positions is key to the success of the method: Sampled too close, they lack information. Outside of the observed are (and thus not represented in $\mathbf{r}_t$), the blind auxiliary agent would have to rely on regularities in environment layouts to navigate, and not on $\mathbf{r}_t$. We sample a large initial number of subgoals and remove less informative ones, ie. those whose geodesic distance $d_G$ to the waypoint is close to its Euclidean distance $d_E$, i.e. $\frac{d_G}{d_E} < T$. For these, following the compass vector would be a sufficient strategy, non-informative about the visual representation. Details are given in Section 5.

**Implementation and downstream tasks** — Akin to training with PPO (Schulman et al., 2017), a rollout buffer is collected with multiple parallel environment interactions, on which the Navigability loss is trained. This facilitates optional batching of PPO with navigability, with both losses being separated over different environments — see Section 5, and the appendix for implementation.

Training of the agent is done in two phases: a first representation training, in which the main policy $\pi$, the representation $\mathbf{r}_t$ and the auxiliary agent $\rho$ are jointly trained minimizing $\mathcal{L}_{Nav}$, eq. (4). This is followed by fine-tuning on the downstream task with PPO. We also propose a combination of Navigability and BC losses using $\mathcal{L} = \mathcal{L}_{Nav} + \mathcal{L}_{BC}$. The advantages are two-fold: (i) training the main policy is not idle for environments selected for the Navigability loss, and (ii) visual observations gathered in environment steps spent in short episodes are not wasted, as they are used for training $\mathbf{r}_t$ through backpropagation through the main policy — see Section 5.

## 4 ON THE IMPORTANCE OF THE BLINDNESS PROPERTY

We argue that blindness of the auxiliary agent is an essential property, which we motivate by considering the impact of the supervised learning objective in terms of compression and vanishing gradients.

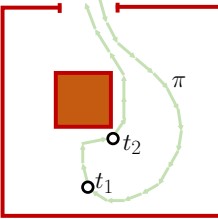

Figure 4: Beh. cloning.

Figure 4 shows a main agent $\pi$ which, after having entered the scene and made a circular motion, has observed the central square-shaped obstacle and the presence of the door it came through. Our goal is to *maximize the amount of information on the obstacles and navigable space extracted by the agent through its training objective*. Without loss of generality we single out the representation estimated at $t{=}t_1$ indicated in Figure 4. While this information is in principle present in the observation history $\{\mathbf{o}_t\}_{t \leq t_1}$, there is no guarantee that it will be kept in the representation $\mathbf{r}_t$ at $t{=}t_1$, as the amount of information storable in the recurrent memory $\mathbf{r}_t$ is much lower than the information observed during the episode. Agent training leads to a learned compression mechanism, where the policy (expressed through equation (5)) compresses $\{\mathbf{o}_t\}_{t \leq t_1}$ in two steps: (1) information from $\mathbf{o}_t$ not useful at all is discarded by $f$ *before* it is integrated into $\mathbf{r_t}$; (2) information from $\mathbf{o}_t$ useful for a single step is integrated into $\mathbf{r}_t$, used by $\pi$ and then discarded by $f$ at the next update, i.e. it does not make it into $\mathbf{r}_{t+1}$. Here we mean by "information content" the mutual information (MI) between $\mathbf{r_t}$ and the observation history, i.e.

$$I(\mathbf{r}_t; \mathbf{o}_{past}) = \mathbb{E}_{p(\mathbf{o}_{past})} \left[ \log \frac{p(\mathbf{r}_t|\mathbf{o}_{past})}{p(\mathbf{r}_t)} \right], \tag{7}$$

where $\mathbf{o}_{past} = \{\mathbf{o}_{t'}\}_{t' \leq t}$. Dong et al. (2020) provide a detailed analysis of information retention and compression in RNNs in terms of MI and the information bottleneck criterion (Bialek et al., 2001).

The question therefore arises, whether the BC objective is sufficient to retain information on the scene structure observed before $t{=}t_1$ in $\mathbf{r}_t$ at $t{=}t_1$. Without loss of generality, let us single out the learning signal at $t{=}t_2$, where $t_2 > t_1$, as in Figure 4. We assume the agent predicts an action $\mathbf{a}_t$, which would lead to a collision with the central obstacle, and receives a supervision GT signal $\mathbf{a}_t^*$, which avoids the collision: ▬↯$\mathbf{a}_t^*$. Minimizing $\mathcal{L}(\mathbf{a}_t, \mathbf{a}_t^*)$ requires learning to predict the correct action $\mathbf{a}_t^*$ given its "input" $\mathbf{r}_t$, and in this case this can happen in two different reasoning modes:

**(r1)** learning a memory-less policy which avoids obstacles visible in its current observation, or
**(r2)** learning a policy which avoids obstacles it detects in its internal latent map, which was integrated over its history of observations.

It is needless to say that (r2) is the desired behavior compatible with our goal stated above. However, if minimizing the BC objective can be realized by both, (r1) and (r2), we argue that training will prioritize learning (r1) and neglect (r2) for essentially two reasons: firstly, the compression mechanism favors (r1) which does not require holding it in an internal state for longer than one step. Secondly, reasoning (r2) happens over multiple hops and requires backpropagation over multiple time instants, necessary for the integration of the observed sequence into a usable latent map. The vanishing gradient problem will make learning (r2) harder than the short chain reasoning (r1).

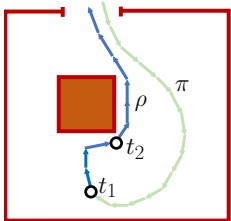

Figure 5: Navigability.

Let's now consider the navigability loss, illustrated in Figure 5. The main agent $\pi$ integrates visual observations over its own trajectory up to waypoint $t=t_1$. The aux agent $\rho$ navigates on the blue trajectory and we again consider the effect of the supervised learning signal at $t=t_2$. Minimizing $\mathcal{L}(\mathbf{a}_t, \mathbf{a}_t^*)$ requires learning an agent which can predict the correct action $\mathbf{a}_t^*$ given its "input" $\mathbf{r}_t$, but now, this can happen in one way only: since the agent $\rho$ is blind, the BC objective cannot lead to reasoning (r1), i.e. memory-less, as it lacks the necessary visual observation to do so. To consistently predict the correct action $\mathbf{a}_t^*$, the representation $\mathbf{r}_t$ collected at $t=t_1$ is necessary, i.e. (r2). Making the aux agent blind has thus the double effect of resisting the compression mechanism in learning, and to force the learning signal through a longer backpropagation chain, both of which help integrating relevant observations into the agent memory. Ribeiro et al. (2020) have recently shown that information retention and vanishing gradients, albeit different concepts, are related.

For these reasons, navigability is different from data augmentation (DA): the steps on short episodes improve the representation integrated over visual observations on long episodes, whereas classical DA would generate new samples and train them with the same loss. We see it as generating a new learning signal for existing samples on long episodes using privileged information from the simulator.

An argument could be made that our stated objective, i.e. to force an agent to learn a latent map, is not necessary if optimizing BC does not naturally lead to it. As counter argument we claim that integrating visual information over time (r2) increases robustness compared to taking decisions from individual observations (r1), in particular in the presence of sim2real gaps. We believe that reactive reasoning (r1) will lead to more likely exploitation of spurious correlations in simulation then mapping (r2), and we will provide evidence for this claim in the sim2real experiments in Section 5.

**Related work** — recent work (Wijmans et al., 2023) has covered experiments on representations learned by a blind agent. Compared to our work, Wijmans et al. (2023) present an interesting set of experiments on the reasoning of a trained blind agent, but it does not propose a new method: no gradients flow from the probed agent to the blind one. In contrast, in our work the blind agent contributes to enrich the representation of a new navigation agent. Our situation corresponds to a non-blind person which is blind-folded and needs to use the previously observed information from memory, with gradients flowing back from the blindfolded situation to the non-blind one.

## 5 EXPERIMENTAL RESULTS

We train all agents in the Habitat simulator (Savva et al., 2019) and the *Gibson dataset* (Xia et al., 2018). We follow the standard train/val split over scenes, i.e. 72 training, 14 for validation, 14 for testing, with approximately 68k, 75k and 71k episodes per scene, respectively.

**Subgoals** — All standard episodes are used as long episodes during training, short episodes have been sampled additionally from the training scenes. To be comparable, evaluations are performed on the standard (long) episodes only. To produce short episodes, we sample waypoints every 3m on each long episode and attach 20 subgoals to each waypoint at a Euclidean distance $\in [3, 5]$m. The threshold for removing uninformative subgoals is set to $T=1.5$m. This leads to the creation of $\sim 36$M short training episodes — no short episode is used for validation or testing.

**Sim2Real** — evaluating sim2real transfer is inherently difficult, as it would optimally require to evaluate all agent variants and ablations on a real physical robot and on a high number of episodes. We opted for a three-way strategy: (i) **Sim2Real** experiments evaluate the model and policy $\pi$ trained in simulation on a real physical robot. It is the only form of evaluation which correctly estimates navigation performance in a real world scenario, but for practical reasons we limit it to 11 episodes in a large (unseen) office environment shown in Fig. 1 and Table 1; (ii) Evaluation in

**Simulation** allows large-scale evaluation on a large number of unseen environments and episodes; (iii) **Sim2NoisySim** allows similar large-scale evaluation and approximates the transfer gap through artificial noise on simulation parameters. We added noise of different types, similar to Anderson et al. (2018), but with slightly different settings: Gaussian Noise of intensity 0.1 on RGB, Redwood noise with D=30 on depth, and actuator noise with intensity 0.5. Details are given in the appendix.

**Metrics** — we report *Success* , which is the number of correctly terminated episodes, and *SPL* (Anderson et al., 2018), Success weighted by the optimality of the navigated path, $SPL = \frac{1}{N} \sum_{i=1}^{N} S_i \left( \frac{\ell_i^*}{\max(\ell_i, \ell_i^*)} \right)$, where $\ell_i$ is the length of the agent's path in episode $i$ and $\ell_i^*$ is the length of the GT path. For robot experiments we also use Soft-SPL (Anderson et al., 2018), which extends SPL by modifying the definition of $S_i$: in failed episodes it is weighted by the distance achieved towards the goal instead of being zero.

**Training setup** — we propose a two-phase strategy with a fixed compute budget of 100M env. steps:

**Phase 1: Pre-training** takes 50M steps. We combine and test 4 strategies: standard PPO, standard BC, Navigability loss and a reconstruction loss (details below), for a total of 8 variants (Table 2).

**Phase 2: Fine-tuning** is the same for all methods, done with PPO (Schulman et al., 2017) for 50M steps on the downstream PointGoal task. We reinitialize the main policy $\pi$ to evaluate the quality of the learned representations, the aux agent is not used. We use a common reward definition (Chattopadhyay et al., 2021) as $r_t = \text{R} \cdot \mathbb{I}_{\text{success}} - \Delta_t^{\text{Geo}} - \lambda$, where $R$=2.5, $\Delta_t^{\text{Geo}}$ is the gain in geodesic distance to the goal, and a slack cost $\lambda$=0.01 encourages efficiency.

Table 1: **Sim2Real transfer** — Avg. performance over 11 episodes in a real environment (Fig. 1 and map below) using the map+plan baseline and three agents trained with PPO, BC and ours, corresponding to variants (a), (c), (e) of Table 2.

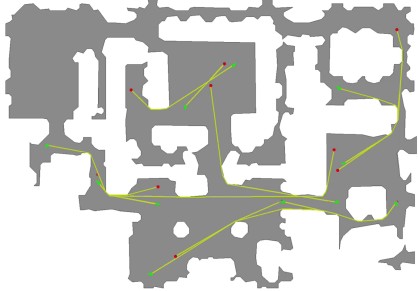

| Method | Success | SPL | sSPL |
|---|---|---|---|
| **(-) Map+Plan** | 36.4 | 29.6 | 32.7 |
| **(a) PPO** | 45.5 | 14.7 | 20.0 |
| **(c) BCⓁⓈ** | 72.7 | 36.3 | 36.3 |
| **(e) Navig.+BC** | **81.8** | **45.6** | **44.0** |

The best checkpoint is chosen on the validation set and all results are reported on the test set.

**Robot experiments** — are performed with a Locobot robot on a large office building with thick carpets, uneven floor and large windows (Fig. 1 and top-down map in Table 1). Results for 11 episodes, shown on the map (avg. GT geodesic length 8.9m), are reported in Table 1 for three end-to-end trained agents (PPO, BC and our proposed agent) and for a map+plan baseline (Gupta et al., 2017) used in (Chaplot et al., 2020b;a), for a total of 44 robot experiments. A detailed description of the agents can be found further below. Overall, the differences in performance are significant, with a clear advantage of the proposed *Navigability* representation in terms of all 3 metrics, providing evidence for its superior robustness (*Success*) and efficiency (*SPL, sSPL*). Qualitatively, the three agents show significantly different behavior on these real robot experiments. Our Navigability+BC variant is more efficient in reaching the goals and goes there more directly. The PPO agent shows zig-zag motion (increasing *SPL* and *sSPL*), and, in particular, often requires turning, whose objective we conjecture is to orient and localize itself better. The BC variant struggled less with zig-zag motion but created significantly longer and more inefficient trajectories than the proposed *Navigability* agent.

**Impact of navigability** — is studied in Table 2, which compares different choices of pre-training in phase 1. For each experiment, agents were trained on 12 parallel environments. These environments where either fully used for training with BC (with a choice of the proposed navigability loss or classical BC of the main policy, or both), or fully used for classical RL with PPO training with PointGoal reward, or mixed (6 for BC and 6 for PPO), indicated in columns *Nr. Envs PPO* and *Nr. BC Envs*, respectively. Agents trained on the navigability loss navigated fewer steps over the long episodes and *saw fewer visual observations, only 5% = 2.5M*. We see that BC (b) outperforms PPO (a), and navigability loss alone (d) is competitive and comparable, outperforming these baselines when transferred to noisy environments. Mixing training with PPO on the PointGoal downstream task in half of the training environments, as done in variants (f) and (g), does not provide gains.

**Optimizing usage of training data** — As the number of environment steps is constant over all agent variants evaluated in Table 2, the advantage of the *Navigability* loss in terms of transferability is slightly counter-balanced by a disadvantage in data usage: adding the short episodes to the training in

Table 2: **Influence of the navigability loss and short episodes**: we compare PPO (Schulman et al., 2017), BC=Behavior cloning and our *Navigability*, which constitute a Phase 1 of 50M env steps. 12 environments are trained in parallel, distributed over the different learning signals. In Phase 2, all of these methods are finetuned with PPO only for an additional 50M env steps. The best validation checkpoints of phase 2 are chosen and evaluated on the test set. "Do Ⓢ" = short episodes are used by the agent. Ⓛ=long episodes, Ⓢ=short episodes. The agent trained with navigability only has seen only 5% of the visual observations (=2.5M).

| Method | Nr. Env steps | Nr. visual obs seen | Do Ⓢ | Nr. Envs $\pi$:PPO | Nr. BC Envs $\pi$:Ⓛ | $\pi$:Ⓢ | $\rho$:Ⓢ | Eval Sim Success | SPL | Eval **Noisy**Sim Success | SPL |
|---|---|---|---|---|---|---|---|---|---|---|---|
| (a) **Pure PPO** | 50M | 50M | – | 12 | – | – | – | 89.6 | 71.7 | 74.6 | 55.8 |
| (b) **Pure BCⓁ** | 50M | 50M | – | – | 12 | – | – | 92.0 | 79.6 | 76.0 | 61.7 |
| (c) **BCⓁⓈ (data augm.)** | 50M | 50M | ✓ | – | 12 | 12 | – | 94.2 | 80.1 | 89.6 | **74.0** |
| (d) **Navig.** | 50M | 2.5M | ✓ | – | – | – | 12 | 92.9 | 77.3 | 86.8 | 68.8 |
| (e) **Navig. + BCⓁ** | 50M | 25+1.25M | ✓ | – | 12 | – | 12 | **95.5** | 80.3 | **90.9** | 73.3 |
| (f) **PPO + Navig.** | 50M | 25+1.25M | ✓ | 6 | – | – | 6 | 91.5 | 72.6 | 85.2 | 63.8 |
| (g) **PPO + Navig. + BCⓈ** | 50M | 50M | ✓ | 6 | 6 | – | 6 | 90.3 | 73.7 | 83.9 | 66.0 |
| (h) **AUX reconst. + BCⓁ** | 50M | 50M | – | – | 12 | – | – | 94.9 | **80.4** | 76.7 | 61.2 |

Table 3: **Impact of the hidden state** of the main policy $\pi$ at transitions between long and short episodes: (c.1) set to zero (short episodes are data augme.); (c.2) set to last waypoint (clear separation of short and long ep.); (c.3) always continue (maximize episode length but introduce sparse teleportations).

| Method | Eval Sim Succ | SPL | Eval **Noisy**Sim Succ | SPL |
|---|---|---|---|---|
| (c.1) **Set to zero** | 90.3 | 74.5 | 85.1 | 64.8 |
| (c.2) **Set to last waypoint** | 88.6 | 74.2 | 81.4 | 63.0 |
| (c.3) **Always continue** | **94.2** | **80.1** | **89.6** | **74.0** |

Table 4: **Communicating with the Mole**: Impact of the choice of connection between representation $r$ and blind policy $\rho$ for agent (e) of Table 2. (e.1) $\mathbf{r}_t$ is fed as "observed" input to $\rho$ at each step; (e.2) as initialization of $\rho$'s own hidden state; (e.3) as previous, but 128 additional dimensions are added to the state of $\rho$.

| Method | $|\mathbf{h}|$ | Eval Sim Success | SPL | Eval **Noisy**Sim Success | SPL |
|---|---|---|---|---|---|
| (e.1) **As observation** | 512 | 92.8 | 77.4 | **90.9** | **73.3** |
| (e.2) **Copy** | 512 | **95.5** | **80.3** | 80.7 | 62.5 |
| (e.3) **Copy+extend** | 640 | 90.4 | 76.9 | 83.3 | 66.6 |

variant (d) has two effects: (i) a decrease in the overall length of episodes and therefore of observed information available to agents; (ii) short episodes are only processed by the blind agent, and this decreases the amount of visual observations available to ∼5%. Combining Navigability with classical BC in agent (e) in Table 2 provides the best performance by a large margin. This corroborates the intuition expressed in Section 3 of the better data exploitation of the hybrid variant.

**Is navigability reduced to data augmentation?** — a control experiment tests whether the gains in performance are obtained by the navigability loss, or by the contribution of additional training data in the form of short episodes, and we again recall, that the number of environment steps is constant over all experiments. Agent (c) performs classical BC on the same data, i.e. long and short episodes. It is outperformed by *Navigability* combined with BC, in particular when subject to the sim2noisy-sim gap, which confirms our intuition of the better transferability of the *Navigability* representation.

**Continuity of the hidden state** — The main agent $\pi$ maintains a hidden state $\mathbf{r}_t$, updated from its previous hidden state $\mathbf{r}_{t-1}$ and the current observation $\mathbf{o}_t$. If this representation is a latent map, then, similar to a classical SLAM algorithm, the state update needs to take into account the agent dynamics to perform a prediction step combined with the integration of the observation. When the agent is trained through BC of the main agent on long and short episodes, as for variants (c) and (e), the main agent follows a given long episode and it is interrupted by short episodes. How should $\mathbf{r}_t$ be updated when the agent "teleports" from the terminal state of a short episode back to the waypoint on the long trajectory? In Table 3 we explore several variants: setting $\mathbf{r}_t$ to zero at waypoints is a clean solution but decreases the effective length of the history of observations seen by the agent. Saving $\mathbf{r}_t$ at waypoints and restoring it after each short episode ensures continuity and keeps the amount of observed scene intact. We lastly explore a variant where the hidden state always continues, maximizing observed information, but leading to discontinuities as the agent is teleported to new locations. Interestingly, this variant performs best, which indicates that data is extremely important. Note that during evaluation, only long episodes are used and no discontinuities are encountered.

**Communication with the Mole** — in Table 4 we explore different ways of communicating the representation $\mathbf{r}_t$ from the main to the aux policy during Phase 1. In variant (e.1), $\rho$ receives $\mathbf{r}_t$ as input at each step; in variants (e.2) and (e.3), the hidden GRU state $\mathbf{h}$ of $\rho$ is initialized as $\mathbf{r}_t$ at the beginning of each short episode, and no observation (other than the subgoal $g_{\cdot}$) is passed to it. Variant (e.2) is the best performing in simulation, and we conjecture that this indicates a likely

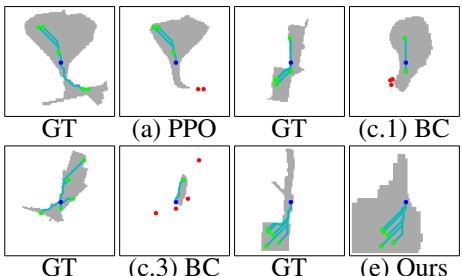

| Method | Sim | | | NoisySim | | |
|---|---|---|---|---|---|---|
| | Reconstr. | 2D Nav. | | Reconstr. | 2D Nav. | |
| | IoU | Succ. | Sym-SPL | IoU | Succ. | Sym-SPL |
| **(a) PPO** | **31.0** | 37.9 | 10.1 | 15.8 | 18.4 | 6.1 |
| **(c.1) BC①Ⓢ** | 25.5 | 45.2 | 11.0 | 11.1 | 16.2 | 4.6 |
| **(c.3) BC①Ⓢ** | 11.9 | 13.2 | 3.5 | 1.1 | 0.6 | 0.3 |
| **(e) Navig.+BC** | 19.8 | **66.5** | **14.7** | **18.0** | **64.8** | **13.2** |

*Viewpoints on figures on the left are not comparable as different agents perform different trajectories. The agent is the blue dot in the center and 5 trajectories out of 10 used for Sym-SPL are plotted. Goal points are green when reachable (always in GT) and red otherwise.*

Figure 6: **Probing reconstruction:** we estimate how much information on reconstruction is contained in the learned representations. PPO (a) appears to capture only parts of the map, leading to failures in this navigation task. Reconstruction by (c.1) exhibits similar characteristics. The (c.3) variant performs poorly on these probing tasks, and the reconstructed map confirms that. Our approach, (e), estimates less accurately than PPO and (c.1) the shape of the navigable space, but still appears to capture important geometric aspects of the environment.

ego-centric nature of the latent visual representation $r_t$. Providing it as initialization allows it to evolve and be updated by the blind agent during navigation. This is further corroborated by the drop in performance of these variants in NoisySim, as the update of an ego-centry representation requires odometry, disturbed by noise in the evaluation, and unobserved during training. Adding 128 dimensions to the blind agent hidden state $h$ in variant (e.3) does not seem to have an impact.

**Probing the representations** — of a blind agent has been previously proposed by Wijmans et al. (2023). We extend the procedure: for each agent, a dataset of pairs $\{(r_i, M_i^*)\}_{i=1..D}$ is generated by sampling rollouts. Here, $r_i$ is the hidden GRU state and $M_i^*$ is an ego-centric 2D metric GT map of size $95 \times 95$ calculated from the simulator using only information observed by the agent. A probe $\phi$ is trained on training scenes to predict $M_i = \phi(r_i)$ minimizing cross-entropy, and tested on val scenes. Results and example reconstructions on test scenes are shown in Fig. 6. We report reconstruction performance measured by IoU on unseen test scenes for variants (a), (c) and (e), with (c) declined into sub-variants (c.1) and (c.3) from Table 3. PPO outperforms the other variants on pure reconstruction in noiseless setting, but this is not necessarily the goal of an actionable representation. We propose a new goal-oriented metric directly measuring the usefulness of the representation in terms of navigation. For each pair $(M_i, M_i^*)$ of predicted and GT maps, we sample $N{=}10$ reachable points $\{p_n\}_{n=1..N}$ on the GT map $M_i^*$. We compute two shortest paths from the agent position to $p_n$: one on the GT map $M_i^*$, $\ell_{i,n}^*$, and one on the predicted map $M_i$, $\ell_{i,n}$. We introduce *Symmetric-SPL* as $Sym\text{-}SPL = \sum_{i=1}^{D} \sum_{n=1}^{N} S_{i,n} \min\left(\frac{\ell_{i,n}}{\ell_{i,n}^*}, \frac{\ell_{i,n}^*}{\ell_{i,n}}\right)$, where, similar to SPL, $S_{i,n}$ denotes success of the episode, but on the *predicted* map $M_i$ and towards $p_n$. Results in Figure 6 show that representations learned with *Navigability* lead to better navigation performance, in particular in NoisySim. While this study is speculative and it is hard do draw conclusive insights from it, these observations seem to corroborate the improved transferability of representations learned with *Navigability*.

**Further comparison with reconstruction** — we claim that unnecessarily accurate reconstruction is sub-optimal in presence of high sim2real gap, and, additional to already discussed robot experiments (Table 1), we compare in simulation with a method supervising reconstruction. The loss is identical to the probing loss described above, but used to train the representation during Phase 1, combined with BC. Corresponding method (h) in Table 2 compares unfavorably with our method (e), in

Table 5: Map+plan baseline.

| Method | Noise | Succ. | SPL |
|---|---|---|---|
| **Class.** | no | **95.7** | **89.9** |
| **Ours (e)** | no | 95.5 | 80.3 |
| **Class.** | yes | 27.9 | 16.3 |
| **Ours (e)** | yes | **90.9** | **73.3** |

particular in noisy settings. In Table 5 we also compare with a classical map+plan baseline of Gupta et al. (2017) and show that under noisy conditions our approach outperforms the classical planner.

## 6 CONCLUSION

Inspired by *Cognitive Maps* in biological agents, we have introduced a new technique, which learns a latent representations from interactions of a blind agent with the environment. We position it between explicit reconstruction, arguably not the desired when a high sim2real gap is present, and pure end-to-end training on a downstream task, which is widely argued to provide a weak learning signal. In experiments on sim2real and sim2noisy-sim evaluations, we have shown that our learned representation is particularly robust to domain shifts.

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

| Method | Steps | Setup | Eval Sim | | Eval NoisySim | | Eval Real | |
|---|---|---|---|---|---|---|---|---|
| | | | Success | SPL | Success | SPL | Success | SPL |
| **DDPO** (Wijmans et al., 2019) | 2.5G | Gibson-2+, SE-ResNetXt101 + 1024-d LSTM | 98.0 | 94.8 | N/A | N/A | N/A | N/A |
| | 2.5G | Gibson-2+, ResNet50 + LSTM-512 | - | 94.0 | N/A | N/A | N/A | N/A |
| **EmbVizNav** (Rosano et al., 2021) | 2.5G+2.4M | Gibson+MP3D+real obs., eval on custom scene, RGB only | 99.1 | 91.3 | N/A | N/A | 97.0 | 80.0 |
| **VizOdomNav** (Partsey & Maksymets, 2021) | 2M | Gibson (Hab. Challenge 2021) | 86.0 | 66.0 | N/A | N/A | N/A | N/A |
| **Robust-Nav** (Chattopadhyay et al., 2021) | 75M+166k | Exhaustive evaluation in sim2noisy-sim | Check paper for a broad study w. multiple params | | | | | |
| **Sadek et al.** (Sadek et al., 2022) | 100M | Real robot, 20 eval episodes only, avg of 2 custom environ. | 86.4 | 71.1 | N/A | N/A | 75.0 | 53.9 |
| **S2R Pred** (Kadian et al., 2020) | 500M | Training w. sliding on, no train noise | N/A | 36.0 | N/A | N/A | N/A | 61.0 |
| | 500M | Training w/o sliding, Training noise | N/A | 36.0 | N/A | N/A | N/A | 61.0 |
| **Ours** | 100M | Hybrid Navigability+BC followed by PPO, no train noise | 95.5 | 80.3 | 90.9 | 73.3 | 81.8 | 45.6 |

Table 6: **Comparison with SOTA** — These numbers are not comparable as no clear experimental protocol exists for sim2real or sim2noisy-sim experiments. The reported experiments feature extreme variations in setups, training regimes, data, number of test episodes. Several papers report a large number of experiments, we reproduce only a small number of variants here. Agent *Ours* is agent (e) in Table 2.

# A  APPENDIX

## A.1  COMPARISON WITH SOTA

Comparisons are difficult, as no clear evaluation protocol exists, in particular when transfer and sim2real performance is targeted. In Table 6 we attempt to provide a non-exhaustive review of PointGoal performance targeting transfer situations particularly. However, we need to point out that the performance numbers are *not comparable*, as there are very large variations in setups, training regimes, data, number of test episodes etc.

## A.2  ARCHITECTURE OF THE AGENTS

An architecture diagram of the full agent is given in Figure 7. The main agent $\pi$ uses a standard architecture from the literature, eg. in (Wijmans et al., 2019) and a large body of follow-up work. A half-width, 4-channels ResNet18 encodes the RGB-D frames into a $512$D vector. A linear layer encodes the pointgoal information with integrated GPS and compass as a $64$D vector. An embedding layer (a trained LUT) encodes the previous action into a $32$D vector. These 3 vectors are then concatenated to form the input of a GRU, which updates the $512$D vector representing the internal state of the agent after each new observation. The internal state is finally fed to 2 linear layers to compute the log-probabilities of each action and an estimate of the current state value, respectively.

When a waypoint is reached along the main path, the main agent stores its internal state (see Section A.5 for implementation details of the training algorithm). The auxiliary agent $\rho$ (a.k.a. the "mole") starts exploring subgoals from the waypoint. Its inputs are the internal state of the main agent when it reached the waypoint, a new "sub-pointgoal" vector pointing towards the currently pursued subgoal, the previously taken action, and its own auxiliary internal state. The "sub-pointgoal" and the previous action are encoded through similar linear and embedding layers as the main agent, respectively. It also has its own GRU, followed by a linear layer producing the action log-probabilities.

As described in the main text and ablated in Table 4 , we tested a few different ways to connect the internal state of the main agent to the auxiliary one: 1. Concatenate it with the other features before feeding it as an input to the auxiliary GRU; 2. Use it to initialize the auxiliary internal state; 3. Use it to initialize part of the auxiliary internal state while keeping some room for values dedicated to the subgoal exploration task.

## A.3  PROBING EXPERIMENTS

### A.3.1  ARCHITECTURE OF THE PROBING AGENT

The probing network's architecture is inspired by the approach proposed in (Wijmans, 2022; Wijmans et al., 2023). It processes the $512$D vector $\mathbf{r}_i$, representing the GRU memory after pre-training (*Phase 1* in Section 5) with a 2-layer MLP with $256$ hidden dimensions to produce an output vector of dimension $1600$. This vector is reshaped into a 3D tensor of size $[64, 5, 5]$ and processed by a Coordinate Convolution (*CoordConv*) layer (Liu et al., 2018), followed by four *CoordConv-CoordUpConv* (Coordinate Up-Convolution) blocks. Each such block is composed of:

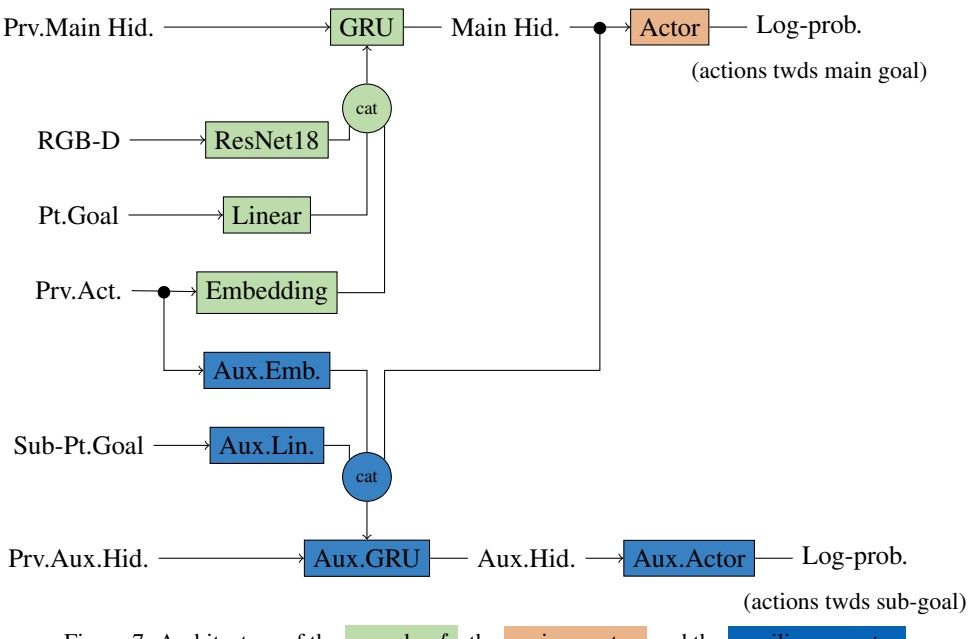

Figure 7: Architecture of the `encoder f`, the `main agent π` and the `auxiliary agent ρ`

**2D Dropout layer** with dropout probability $0.05$;
**CoordConv layer** with kernel size = 3, padding = 1, stride = 1, that reduces the channel size by half while keeping the other dimensions intact;
**CoordUpConv layer** with kernel size = 3, padding = 0, stride = 2, that maintains the channel dimension and doubles the spatial dimensions of the feature map;
**ReLU activation** , except for the last block where it is removed.

This stack produces a feature map of dimension $[4, 95, 95]$ that is processed by a $1 \times 1$-Convolution layer to create the output of size $[2, 95, 95]$, that represents the unnormalized logits of each map pixel being navigable and non-navigable.

For each agent tested, we create a training dataset by running the agent on a subset of 150 episodes for each of the 72 scenes of the Gibson PointGoal training split for a total of 10,800 episodes. We evenly sample each trajectory to obtain approximately 20 observations for each episode, so that we collect training sets of about 200,000 samples. Using the same procedure we also collect a validation dataset from 14 Gibson scenes comprising approximately 20,000 samples, and a test dataset built using the Gibson Pointgoal val split also comprising 20,000 samples.

The probing network is trained with a batch size of $64$ using the AdamW optimizer (Loshchilov & Hutter, 2018) with learning rate $10^{-3}$ and weight decay $10^{-5}$ to minimize the cross-entropy loss $\mathcal{L}_{CE}$ between groundtruth and reconstructed occupancy maps. The validation dataset is used to perform early-stopping, which in probing experiments is particularly important to avoid that scene structure gets its way into the parameters of the probing network.

### A.3.2    ADDITIONAL PROBING EXPERIMENTS

As discussed in the main text, we visualize the probing results for four agent variants: (a), (c) and (e) from Table 1 and Table 2 from the main text, with (c) declined into sub-variants (c.1) and (c.3) in Table 3. The reconstructed maps are shown in Figures 8 and 9. As a reminder, (a) is a standard PPO agent, while (c) variants are trained using behaviour cloning.

Figure 8 shows examples of reconstructed occupancy maps for the four agents in the *Sim* environment without noise. PPO estimates reasonably well the shape of the navigable space, but it is somewhat conservative and captures well only parts of the map, leading to frequent navigation failures. Reconstructions provided by (c.1) exhibit similar characteristics. The (c.3) variant performs poorly according to all quantitative probing metrics, and the reconstructed maps visually confirm that.

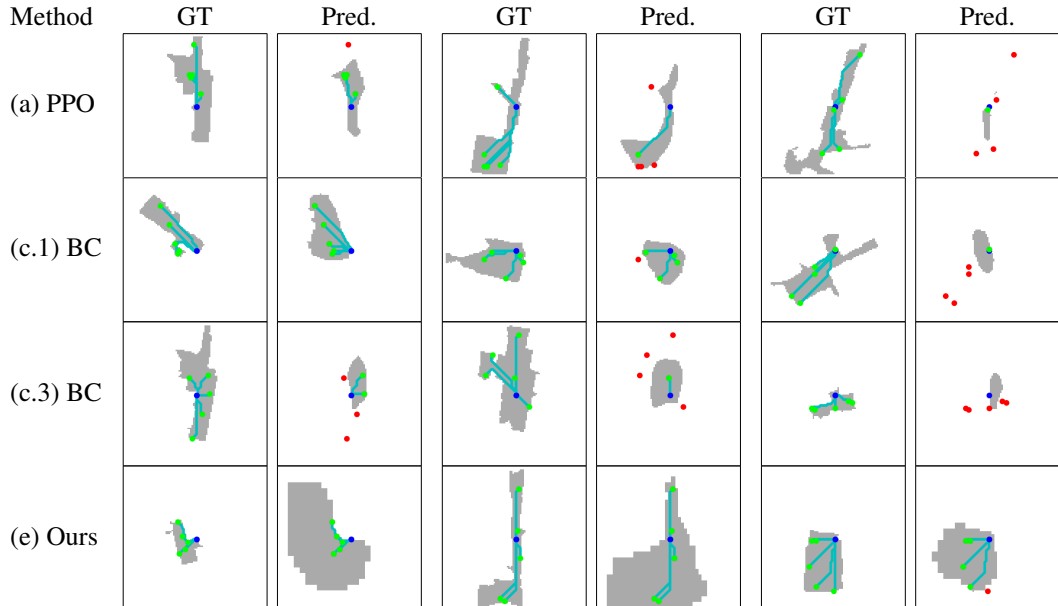

Figure 8: **Additional probing results: *Sim* setting without noise**. Three pairs of groundtruth and predicted navigable maps (in gray) for the four agents (a), (c.1), (c.3) and (e). The agent is in the center (blue dot), and 5 of the 10 trajectories used to compute the Symmetric-SPL are plotted with light blue lines. Trajectories terminate with green goal points when the target is reachable, while target points are red when not reachable, and no trajectory leads to them.

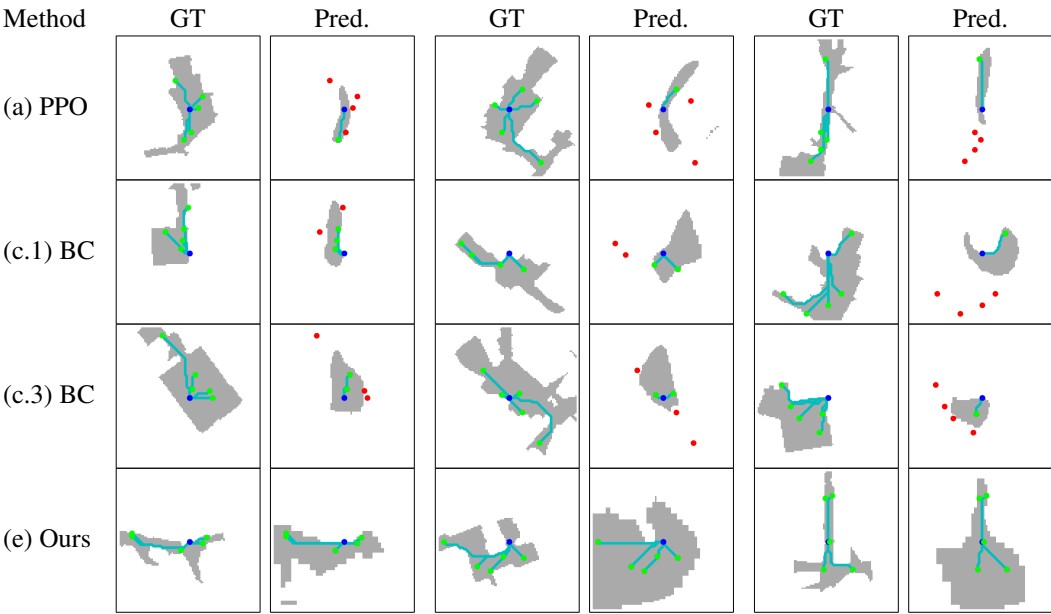

Figure 9: **Additional probing results: *NoisySim* setting**. Three pairs of groundtruth and predicted navigable maps with shortest path trajectories for agents (a), (c.1), (c.3) and (e).

Teleportation seems to severely compromise the representation for the probing task. The proposed approach, agent (e), estimates less accurately than PPO and variant (c.1) the shape of the navigable space, but captures reasonably well the geometry of the environment, especially in the vicinity of the agent.

Figure 9 displays qualitative evaluations on the *NoisySim* environment. In this regime the proposed approach outperforms alternative methods, producing predictions comparable to the noiseless setting, while other methods' predictions degrade significantly.

While the results of the probing experiments only allow to speculate about the nature of the representations learned by the different methods, these observations seem to corroborate that representations learned in the pre-training phase with *Navigability* are more robust and transferable than those of alternative approaches tested in the paper.

### A.4 DETAILS OF THE NOISYSIM EVAL ENVIRONMENT

Complementary to our evaluation experiments on the real Locobot, our evaluation experiments in simulation were conducted in, both, not-noisy and noisy simulated environments. Our noisy environment is largely based on the noise settings of the CVPR Habitat challenge (Kadian et al., 2019), more specifically Gaussian noise on the RGB image, Redwood noise on the Depth image and Proportional Controller noise on the actuators.

However, we have applied some adjustments, motivated by the different objectives of our work, which is to transfer agents from simulation to the target domain with zero domain adaptation to evaluate the transferability of the learned representation. We have observed that in the Habitat implementation of the depth noise model, a depth $D$ above a given threshold $T$ was set to zero[1], i.e.

$$\texttt{if } (D > T) \tag{8}$$
$$D = 0,$$

which is the inverse behavior of the noiseless setting, ie.

$$\texttt{if } (D > T) \tag{9}$$
$$D = T,$$

We argue that this extremely strong discrepancy does not fall into the category of noise but rather to a change in the nature of the sensor, messes up transfer and does not allow a sound evaluation — we therefore replaced (8) by the thresholding process of the standard variant (9).

Secondly, since the original Redwood noise was designed for $640 \times 480$ images (Teichman et al., 2013), applying it to arbitrarily size depth image caused the appearance of black borders[2]. We have extended this by setting depth on the borders to their original values.

### A.5 DETAILS ON THE IMPLEMENTATION OF NAVIGABILITY

Figure 10 illustrates the implementation of the navigability loss in an example showing two long episodes, the first going from $A_1$ to $B_1$, the second from $A_2$ to $B_2$. Each episode has one waypoint each and several sub-goals and short episodes branching out from the waypoints, shown in Figure 10a. As in PPO, learning is implemented with a rollout buffer, which stores a sub sequence of the episode. The rollout buffer of length 128 steps in our implementation can hold one or several long sequences, or only a part of one or more sequences.

As mentioned in the paper, and as is custom in the literature, we train with multiple parallel environments. We strictly separate between two types of environments, and maintain separate rollout buffers for these two types:

**PPO Environments** — are classical environments trained with RL (the PPO variant). As classically done, the rollout buffer is filled in the collection step and used for policy updates in the update step. No short sequences are used, and the trajectories are the ones taken by the agent.

**BC Environments** — are either trained with classical behavior cloning (BC) or with our Navigability loss or with a combination of these two. In this case, as shown in Figure 10b, we flatten the hierarchical structure of long and short episodes into a single sequence, with the agent navigating between waypoints, at each waypoint going through the different short episodes, and then resuming at the waypoint again. The hidden state of the agent is stored at waypoints and restored if necessary, according to the choices ablated in Table 3.

---

[1] https://github.com/facebookresearch/habitat-sim/blob/d3d150c62f7d47c4350dd64d798017b2f47e66a9/habitat_sim/sensors/noise_models/redwood_depth_noise_model.py#L73

[2] https://github.com/facebookresearch/habitat-sim/blob/d3d150c62f7d47c4350dd64d798017b2f47e66a9/habitat_sim/sensors/noise_models/redwood_depth_noise_model.py#L83

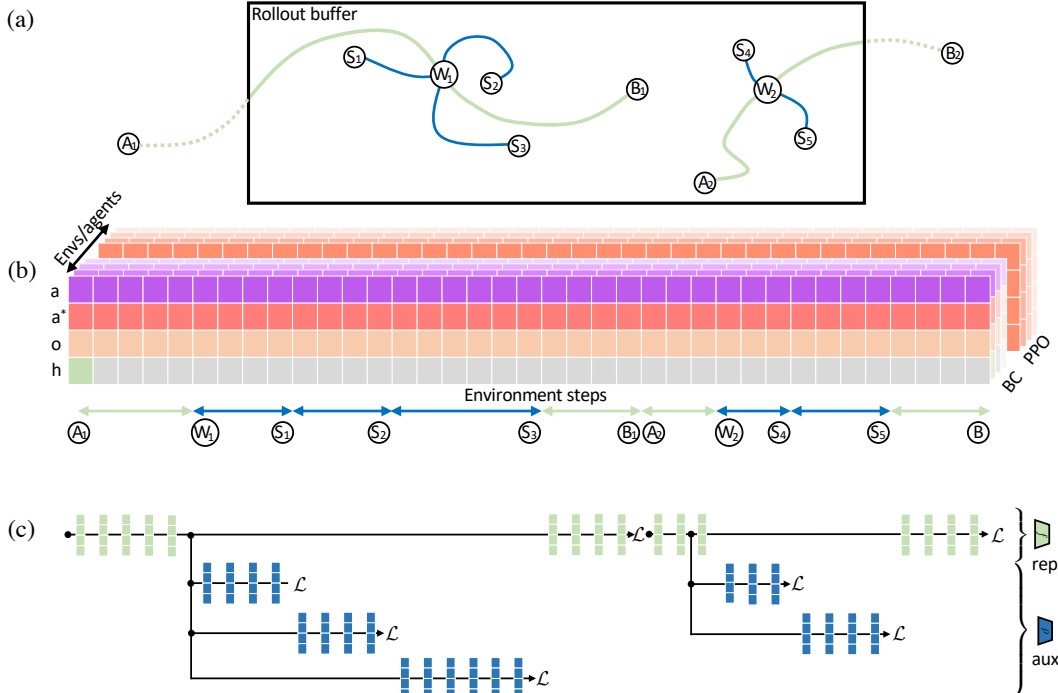

Figure 10: Illustration of the implementation of the navigability loss. (a) An example showing two long episodes, first going from $A_1$ to $B_1$, second from $A_2$ to $B_2$. Each episode has one waypoint each and several sub-goals and short episodes branching out from the waypoints. (b) As in PPO, learning is implemented with a rollout buffer, which stores a sub sequence. (c) The computation graph of an agent only trained on the navigability loss. The hidden state of the representation $f$ is collected on long episodes (Green) and the hidden state of auxiliary agent $\rho$ is collected on short episodes (Blue).

Both types of environments are combined into a single unique batch for gradient updates.

Figure 10c illustrates how the rollout buffer of Figure 10b is mapped to the computation graph of the full agent combining the representation predictor $f$, main policy $\pi$ and the auxiliary policy $\rho$, when everything is trained with the navigability loss only. Vertical time instants of Figures 10a and 10b are aligned and correspond to each other. In its purest form, ie. agent variant (d) of Table 2, only the navigability loss is used for training. In this case, the hidden state of the representation $f$ is collected on long episodes (Green) and the hidden state of auxiliary agent $\rho$ is collected on short episodes (Blue).

## A.6   DETAILS ON MAPPING PREDICTED DISCRETE ACTIONS TO VELOCITY LOCOBOT ACTIONS

In the experiments involving simulation only, all actions were discrete, and as stated in the main paper, were taken from the alphabet {MOVE_FORWARD 25cm, TURN_LEFT, TURN_RIGHT and STOP}. The experiments on the real robot (the Locobot) required mapping the predictions of these discrete actions to velocity commands. A widely used closed-loop strategy executes each action by a closed-loop controller until the discrete action is terminated, then pauses to collect sensor readings, performs a prediction by the neural agent, and repeats. This strategy leads to slow execution and delays between discrete actions.

We have opted for a handcrafted 0-order hold, open-loop velocity command, which led to a speed-up of a factor of around $3\times$-$4\times$. We map each discrete action to an equivalent velocity command $v$ to be sent to the Kobuki driver of the LoCoBot through ROS for execution. Then, we wait for a fixed waiting time $\Delta_t$ to collect the observations (RGB-D and pose). Here, $v$ is either linear velocity $v_l$ or angular velocity $v_a$ depending on the action taken. We empirically, found that setting $v_l$ to 0.25m/sec and $v_a$ to $\pm$ 60deg/sec for a fixed waiting ($\Delta_t$ = 2 seconds) was the best setup to execute the required discrete actions.

