# OpenReview forum: "Learning with a Mole: Transferable latent spatial representations for navigation without reconstruction"
_ICLR.cc/2024/Conference — ICLR 2024 poster_

### Official Review · Reviewer_SbcL · 2023-10-30

**Soundness:** 3 good
**Presentation:** 3 good
**Contribution:** 2 fair
**Rating:** 6
**Confidence:** 4

**Summary:**

This work is about learning an abstract latent representation $\mathbf{r}_t$ for navigation based on end-to-end learning, without a metric map.
The representation is inferred by a recurrent neural network from past RGB-D images and PointGoal-task information.
A policy outputs actions based on this representation.

The main contribution is in how the representation is learned: instead of training it through the main policy directly, the authors use an auxiliary "blind" policy.
In training, the auxiliary policy has to navigate to randomly picked subgoals branching off of the main policy path, but it has to do so using only the last inferred representation $\mathbf{r}_t$ from the main policy loop, with no access to new RGB-D observations.
It is argued that this would force the representation to be better suited for navigation in unseen environments (which is debatable, see my comments about this below).

The approach is trained and evaluated in simulation (Habitat, Gibson scenes), and sim2real transfer is studied on a small scale (11 runs on a real robot).

**Strengths:**

- The method appears sound on the whole, and most design choices are motivated.

- The presentation was easy to follow, but there is some room for improvement (e.g. see comment on missing environment examples below).

- The results indicate the method works in the considered environments, with success rates of ca. 90% in simulation and 80% in sim2real. To the best of my knowledge this is reasonable, compared to the prior art on end-to-end learned navigation.

- Ablations of the main method aspects are included, which is appreciated.

- There is some indication of improved sim2real transfer (e.g. compared to PPO), but there are also limitations of the evaluation (see weaknesses).

**Weaknesses:**

- The number of real-world experiments is rather low (11 episodes). While I can see how three baselines + the method itself add up in terms of workload, with only 11 runs I cannot tell if the sim2real results are statistically significant. I understand that it is unlikely this can be corrected on a short term during the review process, but it is an important point to consider for future revisions.
- Related to the previous point, the paper is currently missing information about the complexity of the environments and the length of the evaluated navigation runs in them. Some qualitative examples in 2D would be nice, particularly of the real office environment. I am primarily concerned about this because navigation success metrics depend very much on a) the geodesic length of the navigation trajectory and b) how maze-like the layout of the scene is. Revealing the complexity of the considered navigation tasks will make the paper stronger.
- I am not certain if the blindness property (a forefront contribution of the paper) is as useful compared to data augmentation:
    - The results seem to support that subgoal spawning is beneficial. However, I am not convinced that the removal of observations in the auxiliary policy matters a lot. Simple behavior cloning + the subgoal data augmentation ((c) in Table 1) performs very well both in Sim and in NoisySim, better than plain navigability training ((d) in Table 1).
    - I understand that the number of RGB-D observations with the proposed method is reduced, but this is a choice. The number of env interactions remains the same, so the comparison is on even terms.
    - In the sim2real experiment, was the BC baseline ((c) in Table2) trained with the data augmentation regime, or regular? I believe sim2real results with BC + subgoal augmentation compared against the proposed method would have been ideal here.
- The main text should specify the exact data fed into the networks clearly, this would make it much easier to assess the applicability of the solution. If I understand appendix A.2 correctly, the representation GRU takes both RGB-D images (not just RGB) and positional data (GPS + compass orientation), whereas the main text only talks about visual observations. This is somewhat misleading, navigation based on RGB data alone is very different from navigation with RGB-D images (from which collision information is easy to extract) with known poses.
- The probing network is not directly related to the policy, so I am skeptical about the interpretation of the probing results at the very end of sec. 5. I think the paper would be better off by highlighting that this investigation is speculative, and should be taken with a grain of salt.

Overall, I find that some of the main claims are stronger than what the experiments show (e.g. sec. 4) and that the evaluation could have been more precise. Note that I understand and can agree with the intuitive arguments in sec. 4, but these need to be supported better by the results. Still, the proposed training scheme does have some merit and the method seems to work, so I am leaning slightly positive in my assessment.

**Questions:**

- In the sim2real experiments, which BC variant was used as a baseline? With augmentation (i.e. subgoals) or without?
- I am not sure I agree with the interpretation of the probing experiments. Sure, they indicate the reconstructed occupancy grids from the navigability representation are more optimistic, but then again can this be interpreted at all? There is no indication that the policy uses the representation in the same way, not to mention that absolute optimism w.r.t. collisions is not a desirable property per se.

---

> ### Author Response · Authors · 2023-11-20
> **Thanks for the constructive and detailed feedback!**
>
> > The number of real-world experiments is rather low (11 episodes). While I can see how three baselines + the method itself add up in terms of workload, with only 11 runs I cannot tell if the sim2real results are statistically significant.
>
> We agree that the more real-world experiments, the better. As rightfully noted by the reviewer, these experiments are time-consuming, and benchmarking four different methods leads to considerable engineering effort. In this paper we conducted 44 robot experiments, and we would like to stress that most papers in this space present significantly fewer experiments, for example:
> - “Object Goal Navigation using Goal-Oriented Semantic Exploration”. NeurIPS 2020: 20 robot experiments.
> - “Success weighted by completion time: A dynamics-aware evaluation criteria for embodied navigation”. IROS 2021: 12 runs, i.e. 2 episodes, each one repeated 3 times, for 2 methods.
> - “Navigating to Objects Specified by Images”. ICCV 2023: 8 experiments.
>
> > The paper is currently missing information about the complexity of the environments and the length of the evaluated navigation runs in them. [...] I am primarily concerned about this because navigation success metrics depend very much on a) the geodesic length of the navigation trajectory and b) how maze-like the layout of the scene is. Revealing the complexity of the considered navigation tasks will make the paper stronger.
>
> Thanks, the revised manuscript features a top-down map of the real environment with the ground-truth test trajectories in Table 1 (previously Table 2, we swapped Table 1 and Table 2 for readability). We also include the average geodesic length of the test episodes, which is 8.9 meters - for comparison Gibson val episodes have an average length of 5.9 meters. One room of the test environment is shown in Fig. 1 and features thick carpets, uneven floor and large windows that disturb the robot sensing.
>
> > I am not certain if the blindness property [...] is as useful compared to data augmentation: [...] I am not convinced that the removal of observations in the auxiliary policy matters a lot. Simple behavior cloning + the subgoal data augmentation ((c) in Table 1) performs very well both in Sim and in NoisySim, better than plain navigability training ((d) in Table 1).
>
> Differences between methods (c) and (d)  are indeed small in simulation (Table 2 in the revised manuscript, previously Table 1). However we argue that the navigability loss improves sim2real transfer because it relies on non-visual navigability information that has reduced sim2real gap compared to RGB-D, used with data augmentation. Thus it is not surprising that performance in simulation is similar for the two approaches, while we observe significant gains in real-world experiments, Table 2 of the revised manuscript (previously Table 1), where Navigability improves Success Rate and SPL with respect to data augmentation, model (c), by about 10%.
>
> > I understand that the number of RGB-D observations with the proposed method is reduced, but this is a choice. The number of env interactions remains the same, so the comparison is on even terms.
>
> This is indeed true, however the navigability loss brings a significant computational advantage at training because the simulator does not have to render RGB-D observations, allowing to speed-up training and reduce hardware and energy requirements.
>
> > In the sim2real experiment, was the BC baseline ((c) in Table2) trained with the data augmentation regime, or regular? I believe sim2real results with BC + subgoal augmentation compared against the proposed method would have been ideal here.
>
> Indeed, the BC baseline in the sim2real experiment (Table 1 in the revised manuscript, previously Table 2) is the one with subgoal augmentation. We modified the notation on the table to "(c) BC (L) (S)" to clarify this.
>
> > The main text should specify the exact data fed into the networks clearly, this would make it much easier to assess the applicability of the solution. If I understand appendix A.2 correctly, the representation GRU takes both RGB-D images (not just RGB) and positional data (GPS + compass orientation), whereas the main text only talks about visual observations.
>
> The inputs are indeed RGB-D and Goal vector, we clarify this in the revised manuscript.
>
> > The probing network is not directly related to the policy, so I am skeptical about the interpretation of the probing results at the very end of sec. 5. I think the paper would be better off by highlighting that this investigation is speculative, and should be taken with a grain of salt.
>
> This was a design choice: the probing experiments are not directly related to the policy, they are designed to probe the representation built in Phase 1 and gain intuitions of what is captured at that stage of training. We agree that we can only speculate about these experiments, this is added to the paper. We also tone down claims related to these experiments.

---

> ### Comment · Reviewer_SbcL · 2023-11-22
>
> Thanks for responding! The top-down map figure and the BC (L) + (S) clarification in the new Table 1 were much needed improvements from my perspective, thanks for addressing them. The rest of the clarifications also help.
>
> My original assessment was already leaning positive, the above raises my confidence in that assessment. My current score would be between 6 and 7.

---

### Official Review · Reviewer_hBzC · 2023-10-31

**Soundness:** 3 good
**Presentation:** 3 good
**Contribution:** 3 good
**Rating:** 8
**Confidence:** 4

**Summary:**

This work presents an approach to learn an actionable representation of the scene without optimizing for the reconstruction objective. The key idea is to use the learned representation to navigate on multiple short sub-episodes without any direct visual observations. The learned representation is optimized by this blind auxiliary agent for navigability and not reconstruction. Extensive experiments in both simulation and real world demonstrate the effectiveness of the proposed approach over several baselines, eg. BC, PPO, Map+Plan, especially in noisy settings.

**Strengths:**

- The learned representation is optimized for its usability in navigation rather than reconstruction objective. This makes it suitable for any downstream task involving navigation.
- The latent cognitive map can be learned via a blind auxiliary agent without the use of direct visual observations.
- The paper is well written. The related work section is quite comprehensive and provides a good overview of the literature. Sec. 4 provides a good discussion on the importance of blindness and differences with data augmentation.
- Extensive experiments (Table 1) in simulation and NoisySim settings show the benefits of Navigability when combined with policy learning approaches, eg. BC and PPO. Sim2real performance (Table 2) on a physical robot shows its effectiveness over BC, PPO & mapping+planning in real world.
- Ablation study on the continuity of hidden state (Table 3), connection between the representation and blind policy (Table 4), and reconstruction performance (Fig. 6) provide valuable insights into the capabilities of the proposed approach.

**Weaknesses:**

- Sec. 4 reasoning makes sense for GRUs & LSTMs since the gating mechanism in GRUs & LSTMs explicitly encourages the specified behavior. Would this also hold for other architectures, eg. transformer? It'd be useful to provide some insights into this.
- Sec. 4 argues that the training will prioritize learning `r1` and neglect `r2` and provides two reasons. Is there any empirical evidence for this in the context of navigation?
- For BC(L)(S)(data augm.) baseline in Table 1, when training on the short routes, is the hidden state copied from the long trajectory or initialized separately?
- How are the c.1 & c.3 variants in Fig. 6 different from Table 3? The c.3 variant in Fig. 6 is worse than other baselines but it works the best in Table 3. It seems like `Always continue` works well with navigability but not with BC. It'd be helpful to clarify why the performances are so different.
- What does `sym-SPL` represent? why is the minimum of the two ratios taken?

**Questions:**

The questions are mentioned in the weaknesses above.

---
I have read the other reviews and author's response and I am retaining my rating. I think the paper will be a good addition to the conference.

---

> ### Author Response · Authors · 2023-11-20
> **Thank you for the constructive and detailed feedback!**
>
> Thank you for the constructive and encouraging remarks!
>
> > Sec. 4 reasoning makes sense for GRUs & LSTMs since the gating mechanism in GRUs & LSTMs explicitly encourages the specified behavior. Would this also hold for other architectures, eg. transformer? It'd be useful to provide some insights into this.
>
> The reasoning in section 4 is not limited to any specific form of representing the history of observations of the agent. It can be compressed into a recurrent memory, as in our experiments (GRU or LSTM), or be self-attention over time, like in a decision transformer.
>
> > Sec. 4 argues that the training will prioritize learning r1 and neglect r2 and provides two reasons. Is there any empirical evidence for this in the context of navigation?
>
> While compression mechanisms in recurrent memory have been studied in the past (cf. the references we provide in section 4.), up to our knowledge, we are the first to investigate them in the context of navigation.
>
> On the other hand, short-cut learning is a well established concept in the literature in a very general context, establishing that, if the training data contains spurious correlations, trained models tend to learn simpler (mostly unwanted) decision strategies:
> - Leon Bottou. “From machine learning to machine reasoning”. Machine learning, 94(2):133–149, 2014
> - Geirhos et al. “Shortcut learning in deep neural networks”. Nature Machine Intelligence, pages 665–673, 2020.
>
> In terms of empirical work specifically on navigation, we can refer to the well known “copy-cat” behavior in autonomous driving, where agents with memory tend to behave less well than agents taking only the current observation, e.g. as in
> - Wen et al., Fighting Copycat Agents in Behavioral Cloning from Observation Histories, NeurIPS 2020.
>
> In the following work it has been empirically shown that an agent attends to the relevant latent memory slot when a searched (and previously seen) object is not in view, and that attention to memory disappears once the object is in view. While this has only been qualitatively shown (Figure 4 and associated explanations) and not quantitatively evaluated, it provides further evidence for the behavior:
> - Beeching et al., EgoMap: Projective mapping and structured egocentric memory for Deep RL. ECML-PKDD, 2020.
>
> > For BC(L)(S)(data augm.) baseline in Table 1, when training on the short routes, is the hidden state copied from the long trajectory or initialized separately?
>
> This is variant (c.3), “always continue”, where the hidden state is copied from the long trajectory.
>
> > How are the c.1 & c.3 variants in Fig. 6 different from Table 3? The c.3 variant in Fig. 6 is worse than other baselines but it works the best in Table 3. It seems like `Always continue` works well with navigability but not with BC. It'd be helpful to clarify why the performances are so different.
>
> The reasoning requirements for agents in these scenarios are different. We can only speculate on the reasons: We expect the “always continue” variant to perform less well because of the inconsistency of the hidden state at the sub goals, and this is corroborated by the experiments in Figure 6, which *only* uses the spatial representation, per construction. However, the full agent presented in Table 3 uses the spatial representation but also reactive components from visual observations, which can learn short-cuts in reasoning. This agent will largely benefit from having more information integrated in its hidden memory, which in this case is not reset at each sub goal.
>
> These are conjectures and it is hard to derive conclusive insights from these probing experiments.  We stress this in the revised paper and we will tone down the claims related to the probing experiments.
>
> > What does `sym-SPL` represent? why is the minimum of the two ratios taken?
>
> sym-SPL is a symmetric version of SPL that we introduce in paragraph “Probing the representations”, page 9: $$\textit{Sym-SPL} = \sum_{i=1}^D \sum_{n=1}^N S_{i,n}
> \min  \left (\frac{\ell_{i,n}}{\ell_{i,n}^*},\frac{\ell_{i,n}^*}{\ell_{i,n}} \right ).$$
> The reason to use this measure is that the reconstructed map used by the agent to navigate might erroneously indicate non-navigable space as navigable, leading to SPL larger than 1. For this reason, besides computing the ratio between the predicted path $\ell_{i,n}$ and ground-truth shortest path $\ell_{i,n}^*$ as in standard SPL, we also compute $\frac{\ell_{i,n}^*}{\ell_{i,n}}$ and take the minimum to penalize planned trajectories that go through non navigable spaces.
>
> In other words, and to give an intuitive explanation, compared to standard SPL, to be optimal, not only needs the agent’s path approach the ground truth (GT) “from below”, it should also not “overshoot”.

---

> > ### Comment · Reviewer_hBzC · 2023-11-22
> >
> > Thank you for the clarifications. It helps me understand the paper better.

---

### Official Review · Reviewer_sYMN · 2023-11-01

**Soundness:** 3 good
**Presentation:** 3 good
**Contribution:** 2 fair
**Rating:** 6
**Confidence:** 3

**Summary:**

The authors address the task or learning an informative latent state representation for downstream tasks. Instead of a conventional scene reconstruction loss they opt to learn a representation that optimizes navigability by introducing a time-series behavior cloning loss for a blind auxiliary policy (with no future timestep observations or reconstructions) on generated short-term sub-goals (section 3-4, page 4).

The authors discuss the importance of the “blindness property” with a toy example in section 4 (Figures 4 and 5) where they explain that the conventional behavior cloning method may learn to adopt a memory-less latent representation whereas their approach involving future timestep predictions based on past observations encourages memory in the latent representation.

The method is evaluated in simulation and real-world experiments. They report raw navigation success rate and SPL (success weighted by the optimality of planned paths) versus variants of behavior cloning and PPO (Table 1 and 2) where they method outperforms baselines. They further examine how to continue the latent state at the end of short sub-goal episodes (Table 3) and how to set the initial latent for the auxiliary blind policy (Table 4). The authors also examine the representative ability of their latent state by training a scene map reconstruction network and doing subsequent path planning with it (Figure 6 and Table 5). Although other baselines obtain a better raw map reconstruction, their method generally outperforms others in the navigation planning score (especially in noisy environments).

**Strengths:**

- Discussion and experiments are detailed and analyze the different components of the method. Experiments test many ablations and variants of baselines to illustrate the impacts of their additions.
- Both simulation and real-world results.
- The work is well written and clear.

**Weaknesses:**

I feel the prediction of future labels without reconstruction is not a very novel architecture. I am also not sure if the improvement in results is because of the “blindness property” in the navigability loss (equation 5) or simply the addition of training data from generated sub-goals. There is an experimental discussion section addressing this concern (“Is navigability reduced to data augmentation?”, page 8) where they compare against a behavior cloning agent with added sub-goals. However, the difference in performance ((c) versus (e) in Table 1) seems relatively small (about 1% for success rate and even smaller point margins for SPL) and there is no reported confidence intervals or variance for these results over repeated seeds or trials. Thus, I am unsure how convincing these results are.

__Minor Criticisms:__
- Table 2 seemed to be discussed in detail before Table 1 in the experimental section which I found a bit disorienting for readability purposes.
- I believe the methods relation to a mole is only explained in appendix  A.2. (The auxiliary agent ρ (a.k.a. the “mole”)) and so I was initially a bit confused by the title.

**Questions:**

- On page 8, “Impact of navigability”, the text reads: “We see that PPO (a) outperforms classical BC (b), which corroborates known findings in the literature.” However, the results in Table 1 appear to indicate the opposite, with BC (b) having a higher success rate. Is this a typo or a misunderstanding on my part?

- In Table 3, the “c.3. Always continue” variant performs best. However, in Figure 6, now c.1 outperforms c.3. Is there any intuition behind this result?

---

> ### Author Response · Authors · 2023-11-20
> **Thank you for your feedback!**
>
> Thanks for the constructive remarks!
>
> > I feel the prediction of future labels without reconstruction is not a very novel architecture. I am also not sure if the improvement in results is because of the “blindness property” in the navigability loss (equation 5) or simply the addition of training data from generated sub-goals.
>
> The main difference between adding the navigability loss and standard data augmentation comes from the fact that the auxiliary agent is blind: the steps on short episodes improve the representation integrated over visual observations on long episodes, whereas classical data augmentation would generate new samples and train them with the same loss. We see this as generating a new learning signal for the existing samples (i.e. on long episodes) using privileged navigability information from the simulator. The experiments with the real robot also clearly show the advantage of this method compared to data augmentation.
>
> > There is an experimental discussion section addressing this concern (“Is navigability reduced to data augmentation?”, page 8) where they compare against a behavior cloning agent with added sub-goals. However, the difference in performance ((c) versus (e) in Table 1) seems relatively small (about 1% for success rate and even smaller point margins for SPL) and there is no reported confidence intervals or variance for these results over repeated seeds or trials. Thus, I am unsure how convincing these results are.
>
> Our hypothesis is that the navigability loss improves sim2real transfer because it relies on non-visual navigability information that has reduced sim2real gap compared to RGB-D, used with data augmentation. Therefore it is not surprising that performance in simulation is similar for the two approaches (Table 2 of the revised manuscript - previously Table 1), while we observe significant gains in real-world experiments (Table 1 of the revised manuscript - previously Table 2), where Navigability (e) improves Success Rate and SPL with respect to (c) by about 10%.
>
> > Table 2 seemed to be discussed in detail before Table 1 in the experimental section which I found a bit disorienting for readability purposes.
>
> Thanks a lot for the comment, in the revised manuscript we swapped Table 1 and Table 2 to improve readability.
>
> > I believe the methods relation to a mole is only explained in appendix A.2. (The auxiliary agent ρ (a.k.a. the “mole”)) and so I was initially a bit confused by the title.
>
> The name “mole” was chosen because moles are blind, as our auxiliary agent is, so we found it to be an evocative image.
>
> > On page 8, “Impact of navigability”, the text reads: “We see that PPO (a) outperforms classical BC (b), which corroborates known findings in the literature.” However, the results in Table 1 appear to indicate the opposite, with BC (b) having a higher success rate. Is this a typo or a misunderstanding on my part?
>
> Thanks a lot for the remark, this was indeed a typo and it is corrected in the revised version of the manuscript.
>
> > In Table 3, the “c.3. Always continue” variant performs best. However, in Figure 6, now c.1 outperforms c.3. Is there any intuition behind this result?
>
> Table 3 shows results for the complete agents on the PointGoal navigation task, e.g. navigation in 3D photorealistic environments, whereas Figure 6 reports results for the experiments probing the representations after Phase 1 of training, e.g. “navigation” in estimated 2D maps. The reasoning requirements for agents in these scenarios are different. We can only speculate on the reasons: We expect the “always continue” variant to perform less well because of the inconsistency of the hidden state at the sub goals, and this is corroborated by the experiments in Figure 6, which *only* uses the spatial representation, per construction. However, the full agent presented in Table 3 uses the spatial representation but also reactive components from visual observations, which can learn short-cuts in reasoning not related to the spatial structure. This agent will largely benefit from having more information integrated in its hidden memory, which in this case is not reset at each sub goal.
>
> These are conjectures and it is hard to derive conclusive insights from these probing experiments.  We stress this in the revised paper and we will tone down the claims related to the probing experiments.

---

> > ### Comment · Reviewer_sYMN · 2023-11-20
> >
> > Thank you for replying to my questions. I have raised my score slightly.

---

> > > ### Author Response · Authors · 2023-11-20
> > >
> > > Thank you very much, we appreciate this.

---

### Author Response · Authors · 2023-11-20
**Thanks for your feedback!**

We would like to thank the reviewers for their constructive feedback, which helped improve the quality of the submission considerably. We are happy that they appreciated the interest of the approach (hBzC,SbcL) optimized for usability (hBzC), the extensive experiments in simulation and real (sYMN,hBzC,SbcL) that demonstrate the effectiveness of the approach (hBzC, SbcL) and  improved sim2real transfer (hBzC,SbcL), the insightful ablation studies (sYMN, hBzC, SbcL), and the clarity of presentation (sYMN, hBzC, SbcL).

We have revised the paper to incorporate the reviewers' suggestions, the main modifications are highlighted in blue in the revised manuscript.

In the following, we individually address each remark by first quoting it and then responding as thoroughly as possible.

---

### Meta-Review · Area_Chair_eFPN · 2023-12-05

**Metareview:**

The paper proposes an approach for training a memory-type representation useful for navigation, using an auxiliary blind navigation agent trained on short "sub episodes". The representation supports navigation and leads to competitive performance both in simulation and when deployed on a physical robot.

The reviewers are positive about the work, pointing out that it is well written, proposes an interesting method that, as shown in comprehensive experiments both in simulation and on a physical robot, performs well compared to relevant baselines. The ablation studies are appreciated too.

On the downside, reviewers raise doubts whether the blindness property is as important as claimed.

Overall, this is a solid paper and I recommend acceptance.

**Justification For Why Not Higher Score:**

Performance gains are not dramatic; reviewers raise doubts whether the blindness property is as important as claimed.

**Justification For Why Not Lower Score:**

the paper is well written, proposes an interesting method that, as shown in comprehensive experiments both in simulation and on a physical robot, performs well compared to relevant baselines. The ablation studies are appreciated too.

---

### Decision · Program_Chairs · 2024-01-16

Accept (poster)